# MiR-144: A New Possible Therapeutic Target and Diagnostic/Prognostic Tool in Cancers

**DOI:** 10.3390/ijms21072578

**Published:** 2020-04-08

**Authors:** Omid Kooshkaki, Zohre Rezaei, Meysam Rahmati, Parviz Vahedi, Afshin Derakhshani, Oronzo Brunetti, Amir Baghbanzadeh, Behzad Mansoori, Nicola Silvestris, Behzad Baradaran

**Affiliations:** 1Student Research Committee, Birjand University of Medical Sciences, Birjand 9717853577, Iran; omidkoshki@gmail.com; 2Department of Immunology, Birjand University of Medical Sciences, Birjand 9717853577, Iran; 3Cellular and Molecular Research Center, Birjand University of Medical Sciences, Birjand 9717853577, Iran; z.rezaie222@gmail.com; 4Department of Biology, University of Sistan and Baluchestan, Zahedan 9816745845, Iran; 5Infectious and Tropical Diseases Research Center, Tabriz University of Medical Sciences, Tabriz 5166/15731, Iran; meisamrahmati004@gmail.com; 6Department of Anatomical Sciences, Maragheh University of Medical Sciences, Maragheh 5165665931, Iran; pa.vahedi48@gmail.com; 7Immunology Research Center, Tabriz University of Medical Sciences, Tabriz 5165665811, Iran; afshin.derakhshani94@gmail.com (A.D.); amirbaghbanzadeh@gmail.com (A.B.); 8Medical Oncology Unit—IRCCS Istituto Tumori “Giovanni Paolo II” of Bari, 70124 Bari, Italy; dr.oronzo.brunetti@tiscali.it; 9Department of Cancer and Inflammation Research, Institute for Molecular Medicine, University of Southern Denmark, 5230 Odense, Denmark; bmansoori@health.sdu.dk; 10Department of Biomedical Sciences and Human Oncology DIMO—University of Bari, 70124 Bari, Italy; 11Department of Immunology, Faculty of Medicine, Tabriz University of Medical Sciences, Tabriz 5166614766, Iran

**Keywords:** cancer, microRNA, miR-144, therapeutic target

## Abstract

MicroRNAs (miRNAs) are small and non-coding RNAs that display aberrant expression in the tissue and plasma of cancer patients when tested in comparison to healthy individuals. In past decades, research data proposed that miRNAs could be diagnostic and prognostic biomarkers in cancer patients. It has been confirmed that miRNAs can act either as oncogenes by silencing tumor inhibitors or as tumor suppressors by targeting oncoproteins. MiR-144s are located in the chromosomal region 17q11.2, which is subject to significant damage in many types of cancers. In this review, we assess the involvement of miR-144s in several cancer types by illustrating the possible target genes that are related to each cancer, and we also briefly describe the clinical applications of miR-144s as a diagnostic and prognostic tool in cancers.

## 1. Introduction

MicroRNAs (miRNAs) are small (20–23 nucleotides) noncoding RNAs that are conserved evolutionarily [1] Currently, over 28,000 identified mature miRNAs do not encode proteins, but they can regulate gene expression of approximately 30% of biological proteins [2]. These epigenetic molecules regulate the expression of various genes that are involved in critical physiological processes, including differentiation, proliferation, and apoptosis by binding to the 3′-untranslated region (3′-UTR) of a mRNA molecule [3]. MiRNAs indicate a new perspective in the study of cancers. More than 50% of miRNA genes were discovered to be located within the genomic regions that are involved in cancers, suggesting their role in human cancer pathogenesis. In pathological conditions, MiRNAs can negatively regulate gene expression and they are associated with tumor growth, apoptosis, invasion, and metastasis. In the past, several studies have indicated the ability of miRNAs in the inhibition of tumors as a critical option for the improvement of cancer therapy [4,5]. Tumor activator miRNAs, called oncomiRs, are overexpressed in various cancers. On the contrary, suppressive tumor miRNAs are underexpressed [6,7,8]. Among several miRNAs assessed, miR-144s seem to have a role in several cancers. These miRNAs are located in the chromosomal region 17q11.2, being significantly destroyed in many types of cancers. The predicted stem-loop structure of miR-144s recognized by the miRBase (http://mirbase.org/) is shown in Figure 1A. The miR-144 hairpin gives rise to the “guide strand” miR-144-5p and the sister “passenger” strand miR-144-3p (Figure 1B).

Several studies have revealed that miR-144s have different target genes, among which are rapamycin, zonula occludens1, SFRP1, and ANO1. By targeting specific genes, MiR-144 acts as a tumor suppressor or oncogene. However, previous studies have demonstrated that miR-144s are significantly dysregulated in cancers, even if its up-regulation or down-regulation in tumors is not yet clarified (Figure 2). In this review, we will determine the role of miR-144s and their targets in different cancers and then provide understanding as to how this miRNA mediates proliferation, migration, and apoptosis in tumors.

## 2. Biogenesis of miRNAs

Initially, miRNAs are transcripted by RNA polymerase II as long primary-miRNA (pri-miRNA) transcripts in the nucleus. A nuclear RNase III Drosha (pri-miRNAs) with the cofactor DGCR8 forming the microprocessor complex, which cleaves primary miRNAs at the base of the hairpin and makes a hairpin structured precursor (pre-miRNA) [9]. The pre-miRNA, with a 2-nt 3′ overhang, because of Drosha’s RNase III activity, is exported to the cytoplasm by Exportin5 (Exp5), which is a RanGTP-dependent double-strand RNA-binding protein [10]. Afterward, Exp5 liberates pre-miRNA by hydrolyzing guanosine triphosphate (GTP) to guanosine diphosphate (GDP). Subsequently, pre-miRNA is cleaved by Dicer, another RNase, to create a duplex of miRNA with an average of ~22 base pairs (bps) with a 5 prime (5p) and a 3 prime (3p) strand [11]. One of the strands will be maintained due to the relevant thermodynamic stability of the two ends [12]. Dicer-binding proteins and other RNAs including PACT and TRBP can simplify miRNA duplexes production and the transfer of mature miRNAs to Ago proteins [13] (Figure 3). In this case, the miR-144-3p arm of this miRNA shows to be functionally relevant and its most abundant arm [14]. As a consequence of its link with RNA-induced silencing complex (RISC), miRNAs act by a 6–8-mer sequence, a seed sequence, binding to the 3′ UTR of mRNA transcripts in synergy with the canonical Watson–Crick base pairing to control target expression [15].

## 3. MiR-144 in Lymphomas

### 3.1. MiR-144 in Non-Hodgkin’s Lymphomas

#### 3.1.1. MiR-144 in Diffuse B Cell Lymphoma

Diffuse large B-cell lymphoma (DLBCL) is the most common class of aggressive non-Hodgkin lymphoma originating from the germinal center. It is a heterogeneous group of diseases with variable outcomes that are characterized by clinical features, origin, molecular characteristics, and mutations [16]. DLBCL is a malignancy of large B cells and its incidence is increasing 3 to 4% each year in the US [17]. DLBCL is categorized in three major molecular forms: germinal center B-cell-like (GCB), activated B-cell-like (ABC), and primary mediastinal large B-cell lymphoma (PMLBCL). Although a part of DLBCL patients can be cured, the mortality rate is still approximately 40%. This indicates an urgent need to develop more efficient therapies. MiRNAs are engaged in several physiologic and pathophysiologic processes of B cells, such as maturation, migration, and class-switching of immunoglobulins and production [18]. Wang H et al. evaluated the role of miR-144 in the proliferation and invasion of DLBCL cells. They screened the microRNA profiles in DLBCL samples and cell lines with qRT-PCR and found significant downregulation of miR-144 in DLBCL tissues and cell lines. Furthermore, they discovered the B-cell lymphoma 6 (BCL6) as a direct target of miR-144 and inversely associated with the miR-144 expression. [19]. In particular, animal studies showed that the deletion of the miR-144 locus in aged mice induces the development of B-cell lymphoma and acute myeloid leukemia. The underlying mechanism is that the downregulation of miR-144 leads to the overexpression of Myc (an oncogene) in DLBCL [20]. Conversely, Montes-Moreno et al. reported the upregulation of miR-144 with miR-222, miR-451, and miR-221 in the ABC-type of DLBCL [21].

#### 3.1.2. MiR-144 in Follicular Lymphoma

Follicular lymphoma (FL) is the most common indolent B-cell lymphoma and the second most frequent non-Hodgkin lymphoma worldwide and, despite being incurable, it has a median overall survival (OS) of 20 years [22]. Treatment options for FL patients, including hematopoietic stem cell transplantation, are still uncertain. The classification of aberrantly expressed miRNAs in FL as compared to healthy people has proposed that these molecules may help as novel clinical diagnostic and prognostic biomarkers [23]. Approximately one-third of FL patients experience periods of disease remission with recurrent relapses and there is no molecular marker to predict the relapse. Malpeli G et al. investigated miRNAs profile in 26 FLs and 12 relapse lymph nodes (rLN) using immunohistochemistry and microarrays as reference to address the issue. They reported the significant upregulation of miR-144 in rLN vs non relapsed FL. Moreover, miR-144 was inversely associated with FOXp3+ cells in FLs. Since rLN showed a higher level of miR-144 and a lower number of Foxp3+ cells, the authors hypothesized that both events might cooperate in supporting the proliferation of FL cells. Additionally, they identified several possible targets of miR-144 in FL patients, including NOTCH1, TGFB1, MTOR, and PTEN [24]. Takei Y et al. also reported the significant downregulation of miR-144 in the bone marrow smears (BM) of FL patients when compared with normal persons [25]. MiR-144 could be a marker used to predict the invasion of FL cells into BM without having to resort to biopsy.

#### 3.1.3. MiR-144 in Splenic Marginal Zone Lymphoma

Splenic marginal zone lymphoma (SMZL) is a non-Hodgkin lymphoma malignancy of mature B-cells that originally involves the spleen, but it can also affect peripheral organs. SMZL is a rare malignancy that accounts for less than 2% of all lymphomas and accounts for less than 1% of non-Hodgkin’s lymphomas [26]. Di Lisio et al. evaluated a set of miRNAs in B cell lymphomas, including SMZL identifying expressed miRNAs in varying types of B- cell lymphomas. This enabled the characterization of lymphoma neoplasms, lymphoma type, cellular origin, and finding possible targets as oncogenes. They reported a differential expression of 26 miRNAs (20 upregulated and six downregulated) in SMZLs. Among them, miR-144 was significantly overexpressed in the SMZL samples. MiR-144 is an erythropoiesis regulator, a conclusion that may be associated with the splenic microenvironment [27]. Additionally, Bouteloup M et al. and Peveling-Oberhag J reported the significant overexpression of miR-144 in SMZL [28,29].

#### 3.1.4. MiR-144 in the Lymphoma of the Primary Central Nervous System

Primary central nervous system lymphoma (PCNSL) is a less common subtype of non-Hodgkin’s lymphoma that has poor prognosis and is typically limited to the brain, eyes, and cerebrospinal fluid without proof of systemic spread [30]. With an incidence rate of 0.44 per 100,000, PCNSL accounts for approximately 2% of all primary CNS tumors [31]. Recently, some studies describe the crucial role of miRNA in brain development and function. Some miRNAs can modulate the blood-brain barrier integrity by binding to the 3′-UTR of mRNAs encoding several primary junctional proteins. Evidence showed that several miRNAs, including miR-144, miR-181a, miR-18a, miR-34c, miR-34a, miR-140, miR-137, miR-148b, and miR-429, which were significantly upregulated in CNS, enhanced the permeability of blood tumor barrier (BTB) and downregulated the expression of junction-related proteins [32]. BTB limits the delivery of the chemotherapeutic agent to brain tumor tissues. Cai H et al. evaluated the role of miR-144 in the regulation of BTB. They deleted taurine upregulated gene 1(TUG1) in human glioma specimens. By using Real-time PCR, luciferase reporter assay, and bioinformatics tools, they also that reported that the deletion of TUG1 via upregulation of miR-144 increased BTB permeability, and down-regulated the expression of the tight junction proteins ZO-1, occludin, and claudin-5. Thus, TUG1 and miR-144 may be beneficial therapeutic targets or they may improve BTB permeability [33]. Conversely, Ma, Yan et al. reported no significant difference in the expression of miR-144-5p between PCNSL patients [34]. On the whole, more practical and molecular investigations will be essential for clarifying the importance of abnormal miR-144 expression in CNS malignant lesions.

## 4. MiR-144 in Leukemia

### 4.1. MiR-144 in Acute Myeloid Leukemia

Acute myeloid leukemia (AML) is a complex clonal malignancy that is defined by immature myeloid cell proliferation and BM failure [35]. The incidence of AML in the United States is 3.5 cases per 100,000 and it is higher in patients over 65 years old [36]. In 2008, WHO classified seven AML subtypes: 1—AML with recurrent genetic abnormalities and with gene mutations; 2—AML with myelodysplasia-related changes; 3—therapy-related myeloid neoplasms; 4—AML not otherwise specified (NOS); 5—myeloid sarcoma; 6—myeloid proliferations related to the Down syndrome; and, 7—blastic plasmacytoid dendritic cell neoplasms [37]. Modern treatment approaches for AML are based on parameters related to patients, such as age and performance status, as well as particular disease subtypes [38]. AML shows abnormal miRNA expression that was diversely expressed in the different subtypes. Sun X et al. evaluated the role of miR-144-3p as a biomarker in AML patients. They assessed the expression of miR-144-3p in the BM and peripheral blood of AML patients and healthy controls. Additionally, they evaluated the level of miR-144-3p in HL-60 cells. The result showed that miR-144-3p was significantly higher in both the peripheral blood and BM of AML patients when compared with the controls and in HL-60 cells. In this study, the downregulation of miR-144-3p, as an oncogene, via the upregulation of NRF2, reduced cell viability and promoted apoptosis [39]. Zhao Q et al. reported the same results. In their study, the result showed that miR-144 had significantly decreased in both the peripheral blood and BM of AML patients when compared with healthy patients in the control, which suggested the peripheral blood miR-144 as a potential novel non-invasive biomarker for AML screening [40]. FMS-like tyrosine kinase 3 (FLT3) is a part of the receptor tyrosine kinases (RTKs) that transmit signals from the cell surface into the cell through signal transduction. Mutations of the FLT3 gene occur in almost 30% of all AML cases, with the internal tandem duplication (ITD) expressing the most common type of FLT3 mutation [41]. FLT3 gene alterations are associated with a form of AML that is known as cytogenetically normal AML (CN-AML) [42].

FLT3/ITD is an adverse prognostic marker in adults aged < 60 years with primary CN-AML. A study by Susan P et al. evaluated FLT3/ITD–associated microRNA-expression signature in CN-AML patients 60 years of age or older. They reported the downregulation of miR-144 and upregulation of FLT3, homeobox genes (MEIS1, PBX3, HOXB3) as a possible target of miR-144 in CN-AML [43]. Additionally, the upregulation of miR-144 via direct targeting and upregulation of c-myc (a proto-oncogene) in K562 cells, sensitized them to imatinib [44]. The advent of new chemotherapeutic agents led to a significant decrease in the remission rate of leukemia. Nevertheless, the recurrence rate remains as high as 30–40%. Extramedullary infiltration (EMI) of AML includes a wide variety of clinically significant phenomena [45]. The acute myeloid leukemia 1 protein/protein ETO (AML1/ETO; A/E) fusion gene is responsible for 15% of AML cases and 15–26.7% of young patients with EMI. A study by Jiang L et al. examined the role of miR-144 in the regulation of AML1/ETO+ leukemia cell migration via the p-ERK/c-Myc/MMP-2 pathway. They reported the significant upregulation of miR-144 via the downregulation of the amyloid precursor protein (APP) regulated EMI in AML patients. The patients with high expression of APP were more prone to developing EMI [46]

### 4.2. MiR-144 in Acute Lymphoblastic Leukemia

Acute lymphoblastic leukemia (ALL) is a multi-factorial malignancy with an incidence of over 6000 new cases per year in the United States. ALL is the second most common type of leukemia in adults and approximately 60% of cases occur at age <20 years [47]. In 75% of cases, ALL originate from precursors of the B-cell lineage; the remaining cases consist of malignant T-cell precursors [48]. The expression profiling of miRNAs in ALL could be used for the classification, setting specific diagnoses, and proposing the prognostic importance of ALL in the future [49]. Jin J et al. investigated the expression and possible targets of miR-144 in ALL patients. They examined the gene expression of miR-144 while using qRT-PCR in both cell lines and leukemic T cells of ALL patients and reported the significant downregulation of miR-144 in samples. Further assays showed that the upregulation of miR-144 via the downregulation of its target human formin-2 (FMN2) suppressed proliferation and cell-cycle transition of ALL cells, whereas the miR-144 downregulation did not affect ALL progress [50]. Additionally, Mavarakis KJ identified the significant upregulation of miR-144, as a tumor suppressor gene, in acute T-ALL and downregulation of its several possible targets, including *FBXW7*, *BIM*, *NF1*, *PTEN*, *IKZF1*, *and PHF6* [51].

### 4.3. MiR-144 in Chronic Myeloid Leukemia

Chronic myeloid leukemia (CML), with more than 6660 newly diagnosed cases in the United States, is responsible for almost 15% of adult leukemias [52]. CML is associated with the Philadelphia chromosome t(9;22)(q34;q11) and the *BCR-ABL1* fusion gene. Immunotherapy with four FDA approved TKIs, including imatinib, nilotinib, dasatinib, and bosutinib, are first-line treatment of CML patients who are newly diagnosed [53]. Recent studies suggested important roles of miRNAs in the origin, pathobiology, progression, and clinical outcome of CML [54]. Liu L et al. evaluated the regulatory feedback between the myc and the miR-144/451 clusters in CML. C-myc is necessary for BCR-ABL CML cell proliferation. They found that the c-myc expression is upregulated in imatinib-resistant K562R cells, which enhances the miR-144/451 expression [55]. Blast crisis (BC) is the sword of Damocles that hangs over every CML patient. CML has three phases: chronic, accelerated, and blast crisis. BC is the result of maintained BCR-ABL activity, leading to genetic instability, DNA damage, and impaired DNA repair [56]. Machova Polakova, K et al. reported decreased levels of four miRNAs, including miR-144 in PB cells from blast crisis [57].

### 4.4. MiR-144 in Chronic Lymphocytic Leukemia

Chronic lymphocytic leukemia (CLL) is a malignancy of CD5^+^ B cells that is characterized by the accumulation of small, mature-appearing lymphocytes in the blood, marrow, and lymphoid tissues [58]. Blood counts, blood smears, and immunophenotyping of circulating B lymphocytes were used as the diagnostic tools of CLL [59]. CLL is one of the most commonly diagnosed leukemia controlled by oncologists. Several dysregulated miRNAs could play roles as oncogenes or tumor suppressors in CLL. Additionally, they can be used as prognostic biomarkers and as targets for novel treatments [60]. Ruiz-Lafuente N et al. evaluated a set of miRNAs that significantly expressed in CLL when compared with normal B cells (NBC). They reported the upregulation of several miRNAs in CLL, including miR-144-5p, miR-144-3p, miR-28-5p, miR-486-5p, and miR-486-3p. These miRNAs were regulated by IL-4 in CLL [61]. Another study by Gao C et al. using bioinformatics analysis identified miRNA-mRNA target pairs in CLL. They reported a set of 34 differentially expressed miRNAs, including 29 upregulated and five downregulated miRNAs. Among these miRNAs, miR-144 and miR-181a were the most downregulated miRNAs. Additionally, the titin (TTN) gene was identified as a target gene of miR-144, and it was upregulated in CLL [62].

## 5. MiR-144 in Gastrointestinal Cancers

### 5.1. MiR-144 in Gastric Cancer

Gastric cancer (GC) is the second most prevalent cause of cancer-related mortalities worldwide with nearly 740,000 deaths annually [63]. MiR-144 significantly inhibited the proliferation, migration, and invasion in GC cells [64]. Akiyoshi S et al. found that the down-regulation of miR-144 leads to ZFX upregulation and it is associated with GC progression [65]. Mushtaq F et al. investigated the expression pattern, biological functions, and underlying molecular mechanisms of miR-144 in GC cells. They transfected a miR-144 mimic into several GC cell lines, including SGC-7901, AGS, and HGC-27, and subsequently examined protein expression, apoptosis, and gene expression of miR-144 and its potential target. The result of this study showed that miR-144, via targeting activating enhancer-binding protein 4 (AP4), inhibits the proliferation and invasion of GC cells [66]. Furthermore, miR-144 by downregulation of MET signaling reduced GC progression that eventually blocks the activation of the Akt pathway [67]. Moreover, miR-144 has an important role in the inhibition of epithelial-to-mesenchymal transition (EMT) and decreased F-actin expression by targeting pre-leukemia transcription factor 3 (PBX3) in GC cells [68]. Ji TT et al. showed the upregulation of long noncoding RNA TUG1 (lncRNA-TUG1) in GC tissues, while, on the other hand, miR-144 was downregulated. By the inhibition of miR-144, LncRNA-TUG1 can develop the progression and the transmission capacity of GC cells [69]. Helicobacter pylori infection causes active gastritis and it is a risk factor for the intestinal forms of GC. A research study by Lario S et al. investigated the circulating-miRNA profile of GC patients who had H. pylori infection. A total of 123 patients were enrolled in the study and quantitative real-time PCR was used to discover miRNAs. The results showed that miR-144-3p, miR-134-5p, and miR-451a were deregulated in GC, but using these miRNAs had a moderate diagnostic value. In the light of these findings, further investigations are required to increase their diagnostic accuracy [70]. Another study by Liu S et al. showed that the low expression of miR-144 could be prognostic biomarkers in GC and patients with lower expression of miR-144 have lower five-year overall survival [71]. MiR-144 has several other targets in GC, including ZFX, FOSB, SUCLA2, LSM14A, HDHD2, cyclooxygenase-2 (COX-2), PIM1, and GSPT1, suggesting its role as a tumor suppressor [68,72,73].

### 5.2. MiR-144 in Colorectal Cancer

Colorectal cancer (CRC) represents the third malignancy for cancer incidence. Metastases are the main cause of mortality in CRC patients. Several studies improve the knowledge of molecular mechanisms of CRC mediatization, even if these are still not well understood [74]. Mir-144 could be one epigenetic regulator of mediatization, since Anoctamin 1 (ANO1), a miR-144 target gene that is located on chromosome 11q13, is involved in various biological processes, including angiogenesis, chemotaxis, and adherence. Moreover, the downregulation of miR-144, through the targeting of ANO1, leads to the activation of the epidermal growth factor receptor (EGFR) or MAP kinase signaling pathways, two of the main driver pathways involved in CRC [75].

C-X C motif chemokine ligand 11 (CXCL11) is another target of miR-144 in CRC. CXCL11 was significantly upregulated in CRC tissues, while miR-144 was downregulated. The upregulation of miR-144 via the downregulation of CXCL11 leads to the inhibition of inflammation and tumorigenesis [76]. The mTOR is a serine/threonine kinase that is involved in several biological processes, such as cellular growth, metabolism, and cytoskeleton regulation. The mTOR is a downstream effector of the PI3K/AKT pathway and it is a target of miR-144 in CRC [77]. Recent studies showed that the down-regulation of miR-144 via mTOR upregulation leads to tumor progression [78].

Other studies demonstrated that miR-144 is overexpressed in the feces of CRC patients and it can be used for CRC screening [79,80]. In a randomized trial, Choi HH et al. investigated the expression of miR-92a and miR-144 as noninvasive biomarkers in stool samples of CRC patients. In this study, the stool samples were collected from 29 patients with CRC and 29 healthy controls, and the expression of miR-144 with other miRNAs, including miR-21, miR-92a, and miR-200c, were measured by real-time PCR. Eventually, the result showed that the mean levels and the sensitivity of miR-144 were significantly different between the CRC group and the control group (p <0.001, sensitivity 89.7%) and miR-144 might be a useful marker for the detection of CRC [81].

## 6. MiR-144 in Pancreatic Cancer

Pancreatic cancer (PC) is now one of the more aggressive cancers with a five-year survival of less than 5% [82]. Several miRNAs could be predictive/prognostic biomarkers and/or therapeutic targets in PC [83].

In an in vitro/ex vivo study, Li J et al. demonstrated that miR-144-3p was reduced in PC tissues and PANC-1 cells [84]. MiR-144-3p expression inhibited cell growth in the S-phase cell-cycle, leading to cell apoptosis in vitro. MiR-144-3p regulated proline-rich protein 11 (PRR11 3′-UTR). When the PC cells were transfected with miR-144-3p, PRR11 decreased, with the upregulation of p-JNK and p-p38, with a key role in cancer progression impairment. Conversely, PRR11 activation reduced miR-144-3p induced apoptosis and cycle arrest in vitro. So far, miR-144-3p induced cell cycle inhibition and apoptosis in PC cell impairing PRR11. Additionally, in another in vitro study, the MiR-144-3p results downregulated in vitro evaluations. MiR-144-3p overexpression reduces PC cell growth, chemotaxis, and metastasis [85]. Liu S et al. demonstrated that MiR-144-3p enhancing could downregulated PC cell migration, proliferation, and invasion by inhibiting the expression of FOSB

So far, enhancing miR-144 could provide a new target against PC, even if more studies are warranted.

## 7. MiR-144 in Hepatocellular Carcinoma

Hepatocellular carcinoma (HCC) is the ninth major cause of cancer deaths in the United States and is the most common form of primary liver malignancy [86]. In liver tumors, miR-144 was significantly reduced when compared to non-tumor tissues. MiR-144 leads to the knockdown of the EGFR/Src/AKT axis by targeting EGFR in mice HCC. Decreased levels of miR-144 are correlated with increases in growth and metastasis of HCC cells [87]. The nuclear factor erythroid 2–related factor 2 (Nrf2) is a resistance regulator of oxidants in cells [88]. MiR-144 triggers Nrf2 mRNA degeneration by targeting the 3′UTR region and it leads to reverse chemoresistance in HCC cell lines [89]. E2F transcription factor 3 (E2F3) belongs to the E2F family and it has two isoforms, E2F3a and E2F3b [90]. Cao et al. displayed that miR-144 suppresses HCC proliferation and metastasis by targeting E2F3 [91]. Another study detected that the levels of miR-141-3p were significantly increased in serum extracellular vehicles (EVs) and liver cancer tissues as compared with serum and the distal liver tissues in HCC patients. Therefore, the investigation of the miR-144 in EVs might present novel biomarkers for the early and accurate diagnosis of HCC and other liver diseases [92]. MiR-141-3p also leads to invasion and metastasis of HCC by the direct targeting of SMAD4 and SGK3 [93,94]. Yu M et al. assessed the role of miR-144 in HCC. They found that the expression of miR-144 was regularly downregulated in human HCC tissues and cell lines, and the overexpression of miR-144 via direct targeting of SMAD4 significantly repressed metastasis, invasion, cell cycle, EMT, and resistance to chemotherapy [95]. Lu et al. proposed that the upregulation of miR-144 through direct targeting of taurine upregulated gene 1 (TUG1) contributed to the inactivation of the JAK2/STAT3 pathway, impairing proliferation, migration, and tumorigenesis in HCC [96].

## 8. MiR-144 in Cholangiocarcinoma

Cholangiocarcinoma (CCA) is the most common biliary tract primary malignancy and the second most common liver malignancy following HCC, worldwide [97]. Asia has more CCA patients than other regions, but the prevalence of CCA is increasing in North America and Europe [98]. CCA tumors are categorized into intrahepatic (iCCA), perihilar (pCCA), and distal (dCCA) subtypes [99]. It is important to recognize new epigenetic biomarkers, such as miRNAs for CCA patients, since CCA is a host-specific malignancy and it is associated with genetic background. Yang R et al. analyzed miRNA profiles in CCA patients. In this study, the gene expression of miRNAs was confirmed in CCA tissues and CCA cell lines by qt-PCR. They found that miR-144 was significantly down-regulated in CCA tissues and suggested that miR-144 via targeting of LIS1 and decrease of AKT pathway activity may be a primary inhibitor of CCA cell proliferation and invasion [100]. Additionally, in a meta-analysis by He Y et al., he low expression of miR-144 and miR-184 was associated with poor prognosis and differences in the overall survival time of CCA patients. The overexpression of miR-144 can prolong patient survival. Additionally, they found the platelet activating factor acetylhydrolase 1b regulatory subunit 1 (PAFAH1B1) as one of the targets of miR-144 in CCA [101].

## 9. MiR-144 in Esophageal Cancer

Esophageal cancer (EC) is a common cancer of the digestive system, which affects more than 450,000 people in the world [102]. Previous studies demonstrated that the down-regulation of miR-144/451 is related to the higher risk of EC [103]. The high expression of miR-144-3p, miR-144-5p, and miR-451 through targeting Myc and P-ERK led to the apoptosis and inhibition of migration, invasion, and proliferation of EC cells [104]. Another study by Sharma P et al. revealed that miR-144 is an oncomiR in EC cells by targeting PUR-aplha (PURA) [105]. They examined the role of miR-144 in EC by silencing it in KYSE-410 cells and followed this with cell cycle analysis and the cell viability and invasion assays, such as MTT, annexin, and matrigel invasion assay. Their result showed that the miR-144 blocked significantly suppressed EC cell proliferation at 72 h post-transfection (*p* = 0.029). Additionally, the inhibition of miR-144 significantly reduced the migration and invasion of KYSE-410 cells as compared to cells that were treated with negative control (NC). In a study, Wu W et al. examined miR-144 expression in the saliva of patients with esophageal cancer. They collected saliva samples from EC patients and 50 healthy people as the control group. qRT-PCR was utilized to detect the expression of miRNA-144 in the samples. The expression of miR-144 in both the whole saliva and saliva supernatant were significantly higher in the EC group than in the control group (*p* < 0.05). In the whole saliva, the sensitivity and specificity of miR-144 were 74.6% and 92.0%, respectively. In saliva supernatant, the sensitivity and specificity of miR-144 were 53.7% and 94.0%, respectively, thus proposing a moderate diagnostic value of miR-144 in the whole saliva and saliva supernatant [106]. Additionally, the overexpression of miR-144 in serum samples of EC patients could function as a non-invasive prognostic biomarker in EC [107].

## 10. MiR-144 in Genitourinary System

### 10.1. MiR-144 in Cervical Cancer

Cervical cancer (CC) is the most common gynecological malignancy and the eighth most common cancer in the world [108]. MiR-144 is significantly decreased in CC cells when compared to normal cervical cells. MiR-144 suppresses the proliferation and angiogenesis of tumor cells by targeting VEGFA and VEGFC [109]. Furthermore, miR-144 overcomes resistance to cisplatin (CDDP) through targeting of LIM homeobox 2 (Lhx2), leading to apoptosis and inhibiting invasion in tumor cells [110]. Wu et al. displayed that miR-144-3p inhibited the growth and metastasis of tumor cells by targeting mitogen-activated protein kinase 6 (MAPK6) [111]. These results showed that miR-144 might function as a new clinical target or potential biomarker with efficacy in the diagnosis and treatment of metastatic CC.

### 10.2. MiR-144 in Ovarian Cancer

Ovarian cancer (OC) is one of the most frequently diagnosed malignant diseases of women worldwide and it causes more than 150,000 women death a year [112]. The expression of miR-144, miR-93, and miR-382 was reduced in primary ovarian tumors. The down-regulation of miR-144 by targeting the oncogene kinesin family member 14 (KIF14) led to transcriptional and epigenetic regulation [113]. Moreover, the downregulation of miR-144 and miR-216 have a critical role in the lymphovascular invasion of OC [114]. Some studies suggested that the therapeutic potential of miRNAs targeting in the regulation of EMT and apoptosis could contribute to the higher risk of ovarian cancer [115,116]. Another study by Penyige A et al. detected several miRNAs with different expressions in OC, including hsa-miR-144-3p. These results showed that miR-144 functions as a possible biomarker in plasma samples of OC patients and it might be a potential therapeutic target for the treatment of OC [117].

### 10.3. MiR-144 in Prostate Cancer

Prostate cancer (PrC) is one of the most common cancers among men and the third major cause of cancer death in men [118,119]. However, the early diagnosis of PC through prostate-specific antigen (PSA) testing remains unproven, using other biomarkers has the benefit of reducing the overdiagnosis related to the PSA screening [120]. The screening of miRNAs, as a new method of detection has revealed controversial results in identifying PrC, but most of them reported the upregulation of miRNAs [118]. Liu F et al. reported that the overexpression of miR-144 by the down-regulation of Beclin-1, an autophagy-associated protein 6, led to an increase in tumor cell death [121]. Moreover, they found that cisplatin, platinol chemotherapy (CDDP) might induce VEGF through miR-144 levels suppression in PrC cells, with cell autophagy inhibition. Another study by Gu H et al. found that miR-124 and miR-144 directly targeted the 3′UTR of PIM1 and, via its downregulation, led to hypoxia-induced autophagy and increased radiosensitivity of PrC [122]. Zheng H et al. determined that miR-144-3 prevents proliferation and leads to cell death in PrC by targeting CEP55 [123]. Walter BA et al. investigated the value of several miRNAs in cases of PrC from 37 patients. They reported the upregulation of several miRNAs, including miR-144, in the high-grade tumors when compared with normal epithelium [118]. Overall, differently expressed miR-144 with other miRNAs has been suggested to have an impact as a diagnostic and prognostic biomarker in PrC

### 10.4. MiR-144 in Renal Cell Carcinoma

Renal cell carcinoma (RCC) accounts for 70–80% of malignancies in the kidney and it is the most lethal neoplasm of the urologic system [124]. The role of miR-144 in embryonic alpha-hemoglobin synthesis and erythroid homeostasis has been recognized in previous studies. Studies showed that both miR-144-5p and miR-144-3p were significantly downregulated in RCC [125]. Xiang C et al. found that the overexpression of miR-144 via targeting mTOR can suppress cell growth and arrest cells in the G1 phase of tumor cells [126]. Furthermore, miR-144-3p leads to the suppression of invasion and migration, by targeting MAP3K8; so far, it can act as a tumor suppressor [127].

Conversely, Xiao et al. showed that the overexpression of miR-144-3p promoted proliferation, metastasis, and sunitinib resistance by targeting AT-rich interactive domain-containing protein 1A (ARID1A) in clear cell renal cell carcinoma (ccRCC). ARID1A is a transcription regulator and ccRCC is the most common and lethal subtype of RCC [128]. Additionally, in a recent survey, the antitumor roles of miR-451a, miR-144-5p, and miR-144-3p were confirmed in RCC [125]. The assays determined that miR-144-5p and miR-144-3p significantly reduced the migration and invasion in RCC cells, proposing these miRNAs behaved as tumor suppressor miRNAs in RCC. Computational analyses recognized a total of 65 possible targets of miR-144-5p in RCC cells. Among them, high expression of FAM64A, F2, TRIP13, ANKRD36, CENPF, NCAPG, CLEC2D, SDC3, and SEMA4B were dramatically connected with poor prognosis (*p* < 0.001). Among them, the expression of SDC3 was directly regulated by miR-144-5p, and its upregulation enhanced RCC cell invasiveness. Lou N et al. reported a significantly higher level of miR-144-3p in 106 ccRCC plasmas as compared with healthy individuals, suggesting the role of miR-144-3p as a novel and unique plasma biomarker for the diagnosis of ccRCC [129]. Overall, studies showed that miR-144 could function as both tumor suppressor and oncogene, depending on the target gene and pathways involved. Further research is needed for clarifying the importance of miR-144 in RCC and evaluating its clinical potential.

### 10.5. MiR-144 in Bladder Cancer

Bladder cancer (BlC) is one of the highly prevalent cancers and it causes 16,000 deaths annually in the US [130]. Carcinogens, including environmental and occupational exposures, are the main cause of BlC [131]. The role of miRNAs in BlC has been investigated in several studies. Guo Y et al. demonstrated that high expression of miR-144 by targeting the enhancer of zeste homolog 2 (EZH2), leading to the regulation of Wnt pathway and inhibiting the proliferation of tumor cells. Therefore, miR-144 acts as a tumor suppressor in bladder cancer [132]. Matsushita R et al. found miR-144-5p acts as a tumor suppressor by directly targeting CCNE1/2 in BlC. This study was conducted to evaluate the functional roles of miR-144-3p and miR-144-5p and their modulation of targets in BlC cells. Their results showed that miR-144-5p, via the direct targeting of CCNE1, CCNE2, CDC25A, and PKMYT1, dramatically repressed the cell proliferation of BlC cells. Also, patients with higher expression of CCNE1 or CCNE2 had lower overall survival rates than those with low expression (*p* = 0.025 and *p* = 0.032). [133]. Tölle A et al. identified several miRNAs in the blood and urine samples of BIC patients. They reported three significant upregulated blood miRNAs (miR-26b-5p, miR-144-5p, and miR-374-5p) in invasive BlC patients when compared with the control group. The sensitivity and specificity of miR-144 as a tumor marker for the detection of BlC were 70% and 82.4%, respectively [134]. Thus, the current results proposed that the overexpression of miR-144 can function as a new clinical target and as a non-invasive diagnostic tool to detect BlC while using whole blood and urine samples.

## 11. MiR-144 in Lung Cancer

Lung cancer, including small cell lung cancer (SCLC) and non-small cell lung cancer (NSCLC), is a leading cause of cancer-related deaths worldwide for both males and females [135]. The role of miR-144 in lung cancer was assessed in several studies. The concentrations of miR-144 duplex (miR-144-5p and miR-144-3p) were significantly reduced in squamous NSCLC tissues as compared to healthy adjacent tissues. It was observed that both miR-144-5p and miR-144-3p had tumor inhibitory effects by targeting several oncogenes in squamous NSCLC including neuronal calcium sensor 1 (NCS1), solute carrier family 44 member 5 (SLC44A5), and myristoylated alanine rich protein kinase C substrate (MARCKS) genes [136]. The upregulation of miR-144 enhances LINC00483 and Homeobox A10 (HOXA10) genes in lung adenocarcinoma, resulting in a tumor suppressor activity through the regulation of EMT [137]. Chen et al. indicated that miR-144 negatively controls TIGAR expression and induces apoptosis and autophagy in Lung Cancer [138]. MiR-144 is detected in lung cancer cells with the expression of upregulated glucose transporter 1 (GLUT1) and enhanced glucose uptake [139]. Moreover, miR144-5p increases the radiosensitivity in NSCLC cells by targeting activating transcription factor 2 (ATF2) [140]. Chen Y et al. evaluated the downregulation of miR-144-3p and its clinical value in NSCLC. They reported lower expression of miR-144-3p in NSCLC tissue when compared with the normal tissue. Additionally, miR-144-3p expression was significantly associated with stage, metastasis, and invasion of tumor cells, suggesting that miR-144-3p could function as a potential tumor biomarker in the prognosis prediction for NSCLC and as a potential clinical target in lung cancer [141].

## 12. MiR-144 in Mesothelioma

Mesothelioma is an aggressive tumor with long latency after asbestos exposure. Mesothelioma is the most common primary malignancy of the pleural, peritoneal, and pericardial cavities; these three types have a median overall survival of 18, eight, and 11 months, respectively [142]. Mesothelioma nearly affects 2000–3000 persons per year in the USA and it has a poor prognosis [143]. The definite diagnosis of mesothelioma is often challenging and complex, and the qualitative analysis of markers in pleural tissue via immunohistochemical test is the gold standard. Therefore, miRNAs and their molecular mechanisms that are associated with mesothelioma development could be potential diagnostic markers in mesothelioma [144]. Guled et al. conducted the first study aimed at assessing the expression of miRNAs in mesothelioma. The authors investigated the gene expression of 723 miRNAs in 17 tissue samples of mesothelioma while using microarrays. The results showed miR-144 with eight other miRNAs, including let-7e, miR-203, miR-340, miR-34a, miR-423, miR-582, miR-7-1, and miR-9 were significantly down-regulated in the tissue samples of mesothelioma patients when compared to normal tissues [145]. Probably, the loss of miR-144 expression via the upregulation of its potential targets might act as an oncogene in mesothelioma. However, further investigations are needed to clarify the exact role of miR-144 in mesothelioma.

## 13. MiR-144 in Breast Cancer

MiR-144 functions as a tumor suppressor in breast cancer via several targets. Pan Y et al. showed that miR-144 targets 3′-UTR of zinc finger E-box-binding homeobox 1 and 2 (ZEB1 and ZEB2), and controls their expression at the transcriptional and translational levels. Additionally, this study showed that the expression of miR-144 can repress the process of EMT in MCF-7 and MDA-MB-231 cell lines [146]. MiR-144 via the downregulation of CEP55 could inhibit proliferation, invasion, migration, and arresting cell cycle and accelerating apoptosis of breast cancer cells [147]. Another study by Yu L et al. found that miR-144 might function as an oncomiR by an increase in the survival rate of breast cancer cells. The overexpression of miR-144 increased the proliferation rate of tumor cells in the MDA-MB-231 cell line. Additionally, the migration and invasion of tumor cell lines of MDA-MB-231 and SKBR3 were increased by elevated miR-144 expression [148]. Kahraman M et al. showed the overexpression of hsa-miR-144-3p and hsa-miR-144-5p in the aggressive subtype triple-negative breast cancer (TNBC). These miRNAs seem to be potential candidates for invasive detection of basal-like TNBC, prediction of progression-free survival (PFS), and therapy monitoring [149,150].

## 14. MiR-144 in Head and Neck Squamous Cell Carcinoma

Head and neck squamous cell carcinoma (HNSCC) is a member of highly aggressive tumors that involve the epithelium numerous anatomical sections, such as the oral cavity [151]. Zhang et al. demonstrated that miR-144-3p, by targeting of ETS-1 and insulin receptor substrate 1 (IRS1) in laryngeal squamous cell carcinoma, led to the inhibition of metastasis and invasion of tumor cells. Moreover, miR-144-3p, by inhibition of E-cadherin, reduces cellular EMT in laryngeal squamous cell carcinoma [152,153]. Salazar-Ruales c et al. investigated salivary miRNAs for the early detection of HNSCC. They reported the overexpression of miR-144 in the whole saliva and saliva supernatant, suggesting the presence of miRNAs in all salivary fractions for the detection of HNCC neoplasms [154].

### 14.1. MiR-144 in Nasopharyngeal Carcinoma

Nasopharyngeal carcinoma (NPC) is a type of head and neck cancer emerging from the epithelial cells in the nasopharynx. The NPC incidence is much higher in Southeast Asia and North Africa when compared to other regions of the world [155]. The downregulation of anti-oncogenes in NPC by miRNAs has been recognized as an important mechanism of tumorigenesis. MiR-144 plays an oncogenic role in NPC tumorigenesis. Zhang et al. and Song et al. reported that the upregulation of miR-144 by the downregulation of tumor suppressor gene phosphatase and tensin homolog (PTEN) mediates cell proliferation, invasion, and metastasis in NPC specimens and cell lines [156,157]. Moreover, the downregulation of miR-144 by triptolide leads to an increase in p85α-PTEN complex and it causes cell cycle arresting in S-phase in human NPC cells [158].

### 14.2. MiR-144 in Thyroid Cancer

Thyroid cancer (TC) represents 1% of all malignancies, but it is the most common malignancy of the endocrine system [159,160]. Guan H et al. found that the expression of miR-144 is significantly lower in TC tissues when compared to normal tissues. Moreover, miR-144 targets ZEB 1 and ZEB2 and inhibits the invasion of tumor cells, which suggests the role of miR-144 as a tumor suppressor in TC [161]. Sun J et al. proposed that the up-regulation of miR-144 could function as a useful prognostic and clinical target in TC. In particular, the transcription factor E2F8 (a member of E2F family) led to the down-regulation of miR-144 with TC cell proliferation [14]. Sun et al. also confirmed that miR-144 suppresses the proliferation of tumor cells by targeting WW domain-containing transcription regulator protein 1 (WWTR1) in papillary TC [162]. Another in vitro study displayed that miR-144-3p is involved in cell cycle progression and EMT. MiR-144 by targeting Paired box gene 8 (PAX8) caused the activation of signaling pathways, including ERK1/2 and Akt, with consequent tumors proliferation [163]. MiR-144 might offer a potential prognostic and therapeutic strategy in TC.

### 14.3. MiR-144 in Glioblastoma

Glioblastoma (GBM) is the most frequent and malignant brain tumor that is associated with extremely poor prognosis [164]. Aberrant miRNAs expressions can be used as a diagnostic and prognostic biomarker for GBM [165]. Recent studies showed the role of miR-144 as a tumor suppressor by targeting tyrosine-protein kinase Met (c-MET). Several studies verified the downregulation of MiR-144-3p in GBM over non-neoplastic brain tissues [166]. MiR-144-3p negatively exerts tumor growth and apoptosis in glioma cells by targeting topoisomerase II alpha (TOP2A) [167]. Frizzled-7 (FZD7) has been identified as an oncogene that is responsible for cancer cell growth; the activation signaled by Wnt. Cheng et al. indicated that miR-144-3p inhibits metastasis by targeting FZD7 [168]. Besides, some studies demonstrated that the upregulation of miR-144 increases the level of Gli1, a protein that mediates the Hedgehog–Gli pathway in GBM [169]. Moreover, the overexpression of miR-144 by targeting of PDK1, TP53-induced glycolysis and apoptosis regulator (TIGAR), and isocitrate dehydrogenase (IDH)1 and IDH2 lead to cell invasion [170]. Liu ZQ et al. reported that miR-144 inhibits gliomas progression and promotes susceptibility to temozolomide (TMZ, an oral chemotherapy drug) via targeting caveolin 2 (CAV2) and fibroblast growth factor 7 (FGF7). These findings indicated miR-144 functioned as a potential target and new therapeutic strategy and prognostic indicator for gliomas [171].

### 14.4. MiR-144 in Melanoma

Melanoma is the most severe type of skin tumor, with an annual diagnostic rate of more than 130,000 new cases of this malignancy worldwide [172]. Recent findings showed a key role of miRNAs in the progress of melanoma. The down-regulation of miR-144 through targeting of SMAD1 has a regulatory effect on cell proliferation and metastasis in melanoma [173]. Furthermore, miR-144 suppresses the proliferation and invasion of melanoma cells by regulating c-Met expression [174].

### 14.5. MiR-144 in Osteosarcoma

Osteosarcomas (OS) is a rare malignant tumor of the bone that is annually diagnosed in nearly 3.5 cases per million people [175]. Low-level expression of miR-144 was significantly associated with distant metastasis and poor prognosis of OS. Namløs HM et al. reported the significant downregulation of miR-451/miR-144 clusters in OS cell lines [176]. Additionally, Zhao M et al. found that the downregulation of miR-144 by targeting of TAGLN led to the proliferation and invasion of OS tumor cells [177]. They found that miR-144 downregulated in OS cell lines and tissue samples and its expression suppressed proliferation and invasion of OS cells. Additionally, the upregulation of TAGLN, as a target of miR-144, is inversely associated with miR-144 expression Rho-associated kinases 1 and 2 (ROCK1 and ROCK2) are other targets of miR-144 that downregulate them by the upregulation of miR-144, and it has an important role in suppressing the proliferation and invasion [178]. MEF2D, Ezrin, and mTOR are other possible targets of miR-144 that inversely upregulate in OS, suggesting the critical role of miR-144 in the suppression of tumor growth and metastasis [179,180,181].

## 15. Conclusions

It has been established that miR-144 plays a crucial role in erythroid homeostasis. Mice deficient of miR-144 have shown to undergo impairment in late erythrocyte maturation, which further leads to splenomegaly, mild anemia, and erythroid hyperplasia. MiR-144 is a conservative miRNA that has been reported to be aberrantly downregulated in several types of cancers, including diffuse B cell lymphoma, chronic myeloid leukemia, acute lymphoblastic leukemia, colorectal cancer, gastric cancer, pancreatic cancer, hepatocellular carcinoma, cholangiocarcinoma, cervical cancer, ovarian cancer, renal cell carcinoma, bladder cancer, non-small cell lung cancer, mesothelioma, head and neck squamous cell carcinoma, thyroid cancer, glioblastoma, melanoma, osteosarcoma, and primary central nervous system lymphoma. On the other hand, it has been shown to upregulate in follicular lymphoma, splenic marginal zone lymphoma, acute myeloid leukemia, chronic lymphocytic leukemia, breast cancer, nasopharyngeal carcinoma, and esophageal cancer. Evidence shows that miR-144 mostly functioned as a tumor suppressor and inhibited cell proliferation, invasion, metastasis, and tumor growth by targeting several genes. MiR-144 is known not only to target genes, but also to target significant parts of pathways, such as Akt/PKB, EGF/EGFR, Wnt, and JACK/STAT signaling pathways. Additionally, miR-144 targets transmembrane proteins, such as zonula occludens1 (ZO-1), ANO1, and transcription factors, such as STAT family and c-Myc. It has significant roles in the control of cell proliferation, migration, cell death, and tumorigenesis. Our findings indicated that miR-144 could function as a potential therapeutic target and as a new prognostic and diagnostic indicator for a variety of cancers, such as gastric and colorectal cancer.

## Figures and Tables

**Figure 1 ijms-21-02578-f001:**
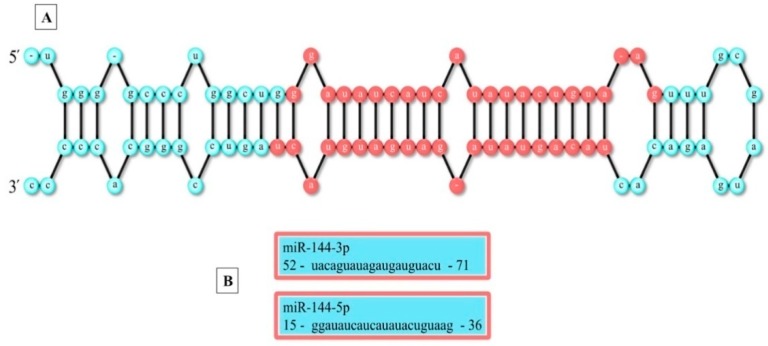
The structure and sequence of miR-144. Note: (**A**) Stem-loop structure of miR-144. (**B**) Two different mature sequences of miR-144.

**Figure 2 ijms-21-02578-f002:**
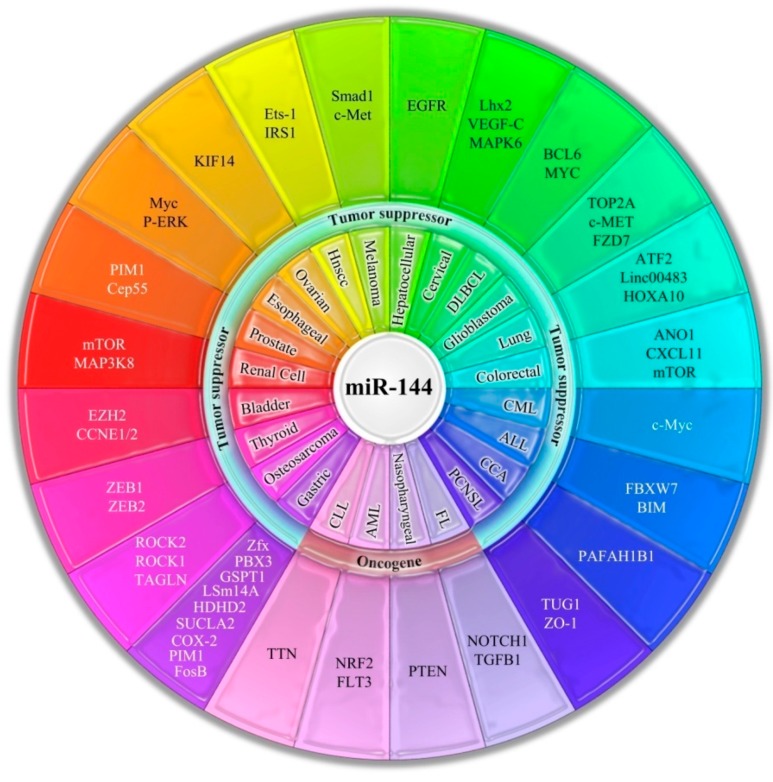
The critical role of miR-144 in different types of cancers. MicroRNAs (miRNAs) are a class of non-coding RNAs that play important roles in regulating gene expression in cancer patients. By targeting specific genes, MiR-144 acts as a tumor suppressor or oncogene. Mostly it downregulated in many types of cancers (see text for detail information). Abbreviations: Diffuse large B-cell lymphoma (DLBCL), Chronic myeloid leukemia (CML), Acute lymphoblastic leukemia (ALL), Cholangiocarcinoma (CCA), Primary central nervous system lymphoma (PCNSL), Follicular lymphoma (FL), Acute myeloid leukemia (AML), and Chronic lymphocytic leukemia (CLL).

**Figure 3 ijms-21-02578-f003:**
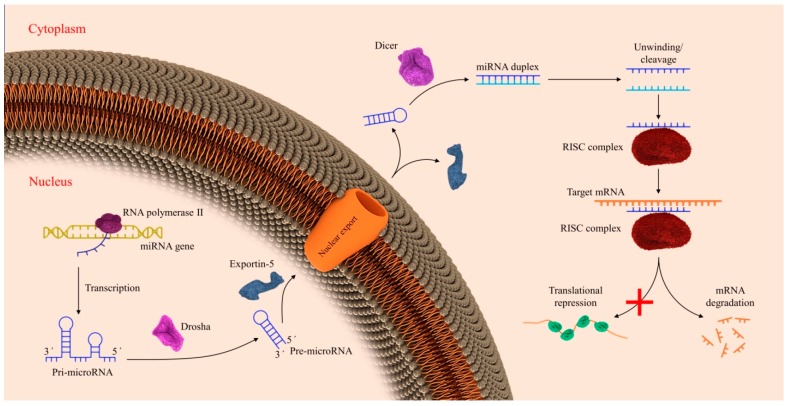
Biosynthesis of miRNAs. A miRNA precursor could be a mature miRNA by various physiological steps (see text for further information).

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
