# Peer review of "MiR-144: A New Possible Therapeutic Target and Diagnostic/Prognostic Tool in Cancers"

_ijms, 2020, doi:10.3390/ijms21072578_

Round 1

Reviewer 1 Report

The manuscript of Omid kooshkaki et al. approaches a very interesting, and, to some extent controversial topic, the role of miR-144 in cancer biology, with a declared focus on its potential as a therapeutic target.

Unfortunately, neither the content, nor the style recommend this work for publication in IJMS.

First of all, I urge the authors to seek help with the writing of this manuscript; not only the wording is at times dubious, but the grammar itself suffers.

Second, the authors fail to properly organize their references and, even more upsetting, fail to cite relevant work in the field! MiR-144 in gastric cancer (probably the smallest chapter) achieves the unpleasant performance to have two identical references (17 and 21), to miss-reference Li B et al (2017) (reference 20; instead, one can find Peng's work on melanoma) and sentenced to oblivion no less than 8 papers:

  1. Li H et al, 2019 (beautiful work on FTX/miR-144/ZFX axis)
  2. Mushtaq F et al, 2019 (miR-144/AP4 interaction)
  3. Lario et al, 2018 (circulating miR-144)
  4. Liu S et al, 2017 (prognostic value of miR-144, correlating serum/tissue expression)
  5. Li B et al, 2017 (which should have been ref 20!)
  6. Ji TT et al, 2016 (miR-144/cMET axis and TUG1 expression)
  7. Li CY et al, 2017 (interesting bioinformatics analysis showing miR-144 connection with overall survival)
  8. Tsai KW et al, 2015 (miR-144 and endothelin-1)

Third, the authors focused only on SOME of the solid tumors (data on thyroid neoplasms, cholangiocarcinomas, mesotheliomas are missing) and no perspective is offered on miR-144 association with lymphomas/leukemias.

Besides the rather unstructured enumeration of miR-144 targets in various types of cancer, there is almost no critical discussion of the referenced data, while essential details (prognostic value, correlation with staging, etc) are sometimes ignored. Furthermore, the biomarker perspective and the discrepancies/concordances between circulating/tissue expression are not consistently/systematically reported and discussed.   

Last but not least, the authors have no comment on the  fact that some papers report on miR-144-3p dysregulation, while others on miR-144-5p (just one of the controversies in the field).

Author Response

We did several modifications to the initial manuscript based on that suggestion of reviewer 1 that highlighted in the revised version of the manuscript and we hope the newer version shows more focus to the goal.

-The authors fail to properly organize their references and, even more upsetting, fail to cite relevant work in the field! MiR-144 in gastric cancer (probably the smallest chapter) achieves the unpleasant performance to have two identical references (17 and 21), to miss-reference Li B et al (2017) (reference 20; instead, one can find Peng's work on melanoma) and sentenced to oblivion no less than 8 papers:

  1. Li H et al, 2019 (beautiful work on FTX/miR-144/ZFX axis)
  2. Mushtaq F et al, 2019 (miR-144/AP4 interaction)
  3. Lario et al, 2018 (circulating miR-144)
  4. Liu S et al, 2017 (prognostic value of miR-144, correlating serum/tissue expression)
  5. Li B et al, 2017 (which should have been ref 20!)
  6. Ji TT et al, 2016 (miR-144/cMET axis and TUG1 expression)
  7. Li CY et al, 2017 (interesting bioinformatics analysis showing miR-144 connection with overall survival)
  8. Tsai KW et al, 2015 (miR-144 and endothelin-1)

Thank you for pointing out that mistake. The suggested references were thoroughly used and the section of gastric cancer was revised and organized.

-Third, the authors focused only on SOME of the solid tumors (data on thyroid neoplasms, cholangiocarcinomas, and mesotheliomas are missing) and no perspective is offered on miR-144 association with lymphomas/leukemias.

Thank you for pointing outthemissing cancers especially lymphomas/leukemias. The new data about the role and effect of miR-144 in missing solid tumors such as cholangiocarcinoma and mesotheliomas and the association of miR-144 with lymphomas/leukemias were added in the newer version of the manuscript.

-Besides the rather unstructured enumeration of miR-144 targets in various types of cancer, there is almost no critical discussion of the referenced data, while essential details (prognostic value, correlation with staging, etc) are sometimes ignored. Furthermore, the biomarker perspective and the discrepancies/concordances between circulating/tissue expression are not consistently/systematically reported and discussed.   

This issue has been discussed in the revised version of the manuscript. The result of our investigation showed that miR-144 might function as a new clinical target or potential biomarker with utility in the diagnosis and prognosis of several types of cancers.

-Last but not least, the authors have no comment on the fact that some papers report on miR-144-3p dysregulation, while others on miR-144-5p (just one of the controversies in the field).

This issue has been discussed in the revised version of the manuscript

Reviewer 2 Report

Kooshkaki et al. have written a nice review on the biology of miR-144. Much is still to be learned about the miRNAs and their biology and this well written review in many ways gives a good overview of our current knowledge about miR-144. The introduction on miRNA biogenesis is well written and to the point. The follow a care review of the cancers in which it has been found and a short review of the data on the effects of either strand (miR-144 or miR-144-3p). Last there is a short summary.

Major points

I think the summary better need to condensate all the information presented on both miR-144 and miR-144-3p. It would be very helpful if the authors could highlight what is common about the targets found in different cancers. Do the act in the same pathway? Which biological processes do they regulate? What are if any the commonalities? Without this type of overview, the review becomes more like a long list of cancers and targets of miR-144/miR-144-3p. However, if added it will greatly improve the paper and its value to its readers.

Minor points

Line 49 and 310: Rapamycin is a drug, not a target of miR-144. Do the authors mean mTOR (mammalian Target of Rapamycin) -please clarify and correct if needed.

The legends to figure are missing. They should be added to make the figures easier to understand.

Figure 2 is missing. If the authors want to have in the paper, it should be added.

Author Response

We would like to thank the reviewer for the positive evaluation of the work and encouragements.

-I think the summary better need to condensate all the information presented on both miR-144 and miR-144-3p. It would be very helpful if the authors could highlight what is common about the targets found in different cancers. Do the act in the same pathway? Which biological processes do they regulate? What are if any the commonalities? Without this type of overview, the review becomes more like a long list of cancers and targets of miR-144/miR-144-3p. However, if added it will greatly improve the paper and its value to its readers.

The summary and the conclusion sections were revised. MiR-144 is known not only to target genes but also to target significant parts of signaling pathways such as Akt/PKB, EGF/EGFR, Wnt, and JAK/STAT pathways. Also, miR-144 targets transmembrane proteins such as zonula occludens1 (ZO-1), ANO1, and transcription factors such as STAT family and c-Myc and playing important roles in the control of cell proliferation, migration, cell death, and tumorigenesis.In the human genome, miR‐451a, miR‐144‐5p (passenger strand), and miR‐144‐3p (guide strand) reside in clustered microRNA (miRNA) sequences located within the 17q11.2 region.We found that miR-144‐5p and miR‐145‐3p act as antitumor miRNAs through their targeting of oncogenes in several cancers.The result of our investigation showed that miR-144 might function as a new clinical target or potential biomarker with utility in the diagnosis and prognosis of several types of cancers. The mTOR is a serine/threonine kinasethatinvolved in several biological processes such as cellular growth, metabolism, and cytoskeleton regulation. The mTOR is a downstream effector of the PI3K/AKT pathway and is a target of miR-144.Recent studies showed that the down-regulation of miR-144 via mTOR upregulation leads to tumor progression

-Line 49 and 310: Rapamycin is a drug, not a target of miR-144. Do the authors mean mTOR (mammalian Target of Rapamycin) -please clarify and correct if needed.

Thank you for pointing out the issue. The sentence was revised.The mTOR is a serine/threonine kinasethatinvolved in several biological processes such as cellular growth, metabolism, and cytoskeleton regulation. The mTOR is a downstream effector of the PI3K/AKT pathway and is a target of miR-144.Recent studies showed that the down-regulation of miR-144 via mTOR upregulation leads to tumor progression.

-The legends to figure are missing. They should be added to make the figures easier to understand.

Thank you for pointing out that mistake. The legends were added in the revised manuscript.

-Figure 2 is missing. If the authors want to have in the paper, it should be added.

Figure 2 was completed and added to the revised manuscript.

Round 2

Reviewer 1 Report

The style of the manuscript is still (colorfully) wordy and at times confusing; however, this version is a significant improvement of the previous submission: the authors did a much better job at collecting the relevant data on miR-144 association with cancer.

It is unfortunate that the manuscript still lacks the logical structuring and the critical view of the contradictory data in the field (mentioning side by side two papers is by no means a critical discussion) much sought for in a good review.

However, should the authors comb again their manuscript for typos and verb/predicate-less sentences, in the present form the manuscript might be considered for publication.

Author Response

Dear Editor,

we tried to improve the text according to the indications of the reviewer